# Using SERVQUAL Method to Assess Tourist Service Quality by the Example of the Silesian Museum Established on the Post-Mining Area

**Natalia Kowalska *** and **Anna Ostręga**

Faculty of Mining and Geoengineering, AGH University of Science and Technology, 30-059 Kraków, Poland; ostrega@agh.edu.pl
* Correspondence: nkowalska@agh.edu.pl; Tel.: +48-537-030-635

**Abstract:** The increasing role of the tourism industry in the global economy and the growing competition makes it necessary to ensure constant performance and continually improve quality. The paper draws attention to the necessity of conducting research on tourist attraction quality also in post-industrial areas which have become attractive tourist sites. It is emphasised that industrial tourism is a new yet quickly developing phenomenon in Poland, which compels managers to differentiate their service range and improve quality standards. The paper employs the SERVQUAL (SERvice QUALity) method to assess the quality of tourist services as a theoretical instrument to measure overall visitor satisfaction. The subject of the research was the Silesian Museum, which is result of reclamation and revitalisation of the inactive "Katowice" Hard Coal Mine. The article presents an empirical verification of the methodology which was modified for the purposes of the research subject. The Museum is considered to be one of the most important cultural centres and a crucial element of the social life of the Silesia region where mining activity has been carried out for centuries. Thirty young people from different continents participated in the survey. The results demonstrated that the expectations of the visitors were not met in three cases only which suggests a very high quality of the Silesian Museum. Moreover, the presented results show increasing capabilities and opportunities to maintain a high quality of services in the studied facility. The findings indicate that the appropriately modified SERVQUAL methodology is a valuable and simple tool to evaluate visitor satisfaction. The results of the evaluation of the Silesian Museum services will be presented to the facility managers. Further research will be carried out after the improvement and implementation of the next stage of the Museum's development.

**Keywords:** post-mining land use; industrial tourism; SERVQUAL method

## 1. Introduction

The expansion and diversification of the tourism industry make it the fastest growing sector of the global economy [1]. The Annual Report of the World Tourism Organization indicates that tourism increased in 2018 by 6% over 2017 [2]. Despite this, the novel coronavirus (COVID-19) has introduced some global challenges in 2020, during which over 90% of the world population was affected by international travel bans [3]. However, due to the highly competitive market, managers of tourist attractions must face the growing requirements of tourists and visitors [4]. This is especially true of domestic tourism on which the tourist traffic in 2020 is currently based. More and more demanding tourists are expected to shape approaches to offerings. Those tourist attractions which do not stand out will give way to those characterised by greater authenticity, educational value, emotional experience, and high standards. In this context, brownfield sites and the right approach to their reclamation and

revitalisation might play an important role. The right approach is understood as the identification, selection, protection, and adaptation of the most valuable industrial heritage. Industrial tourism is breaking popularity records [5]. The technical monuments and post-industrial sites are also becoming more and more often found in the tourist offers in Poland. An increased interest in industrial tourism is noticeable not only among specialists in various fields of technology but also among tourists who want to learn about the broadly understood cultural heritage. The growing importance of industrial tourism is reflected in planning and strategic documents at every level, including European Commission actions in the areas of cultural tourism. The Polish Ministry of Sport and Tourism emphasised the role of industrial tourism in the Tourism Development Programme in response to the demand for products offering new and unique experiences of high educational value. Moreover, it is one of the priority product areas of the Polish tourism industry [6].

The history of the mining industry is illustrated by characteristic spatial structures that dominate the urban landscape, as well as by the machinery and engineering processes which sometimes neglect groundbreaking discoveries. For this reason, in the revitalisation process it is important to preserve the site's authenticity, not only for scholarly purposes, but also to strengthen its identity. The value of brownfield sites for tourism has been recognised for centuries. Once important European mining and metallurgical regions now offer tourist trails with various historic industrial sites. For instance, the Route of Industrial Culture in the Ruhr is visited annually by more than 7.2 million tourists, and the much "younger" Industrial Monuments Route by nearly 1.2 million (data for 2018) [7,8]. The supplement of this route is a tourist route among mining shafts issued in the form of a guide, covering also objects behind the southern border of Poland [9]. The growing interest in industrial heritage brings about the need to ensure greater diversity and improvement of tourist offerings in order to effectively develop this sector of the economy. The reuse of post-industrial land requires skilful management due to the complexity of the process, but it is in line with the principles of circular economy. The aim of the study is to verify the SERVQUAL method as a tool for assessing the quality of tourist attractions and to assess Silesian Museum, which is subject of the study. The museum was created as a result of the revitalisation of a part of the mining area of the "Katowice" Hard Coal Mine in Katowice. The method was modified to adapt it to the nature of the subject of the study which is directly related to the area that has been determined by the heavy industry for many centuries.

Firstly, this paper provides the general context, then the aim and method of the study are presented. Next, it investigates the current state of knowledge, identifying some questions and gaps. Moreover, it presents the research motivation, and finally, the main part of the study includes the results, discussion, and conclusions.

## 2. Research Status Analysis

The effective development of tourism requires the knowledge of the tourist expectations and opinions. Ghose [10] emphasises the significance of testing the quality of tourist attractions by surveying satisfaction and analysing visitors' expectations and experiences. Nowacki [11] points out that satisfaction with a tourist attraction determines one's decision about revisits. In addition, satisfaction builds the long-term success of tourism companies [12]. However, Kruczek et al. [13] indicate that the research into service quality does not translate into the development of marketing strategies by Polish service industry enterprises. In the case of tourism services, quality management is still treated as a marginal activity, and despite understanding the importance of this issue, there is a lack of knowledge in terms of specific actions which could be taken in regard to it [14]. On the other hand, it is quality that shapes the tourist brand [15], and the continuous improvement of the quality of the offered services requires its regular evaluation. Identification of proper techniques to do so in some industries could enhance service quality perception and enable their development and later application of such techniques for the purposes of other industries [16]. In the reference books, many methods of (tourist) service quality assessment are to be found, including SERVPERF [17],

PN-EN ISO 9001:2015-10 Quality management systems [18], Critical Incident Technique (CIT) [19], and the SERVQUAL method [20].

This analysis focuses on SERVQUAL method. The main advantages of this method include universality (no limitations as to the industry or size of an enterprise) and ease of use [21]. Furthermore, it is the only service quality assessment method that allows to obtain information on the discrepancies between the expectations and the actual quality of the services [22]. The SERVQUAL method is a comprehensive construction that can be modified as regards the studied subject, taking into account the local conditions and nature of the offered services. The practical application of SERVQUAL serves as the basis for its development and modification, and can improve the model proposed by Parasuraman. Consequently, it will translate into more reliable results and its more common use in respect of product and tourist attraction management.

The SERVQUAL method was adjusted and validated by various types of service industry quality measurements, including the tourist sector. In general, the literature on the subject indicates the examples of the method's application in many earlier studies on resort towns [23], hotels [24,25], tourist shopping [26], outbound guided package tours [27], tourists' foreign health service [28], medical tourism [29,30], satisfaction levels of two nationalities visiting the same destination [31], casinos [32], airline baggage handling systems [33], city tourism [34], or wine cellars [35]. Meanwhile, the research using the SERVQUAL method was also conducted with regard to museums. A review of the literature revealed that most of the research investigating the service quality in museums show that visitors' expectations are usually met, and their perceptions, and level of satisfaction are evaluated positively. Nowacki [36] assessed the model in the Polish National Museum, identifying that the highest expectations concerned toilets and catering which usually are elements of secondary importance. Moreover, Lau Pei Mey et al. [37] employed SERVPERF in a pilot study at the museum in Malaysia and stated that tested items were within the accepted range except for "the museum charges reasonable entrance fee", which was showed a large standard deviation. In addition, the test identified that tangible factors were less important while experience factors were indeed a major determinant of their service quality expectations. The National Museum in Taiwan has been examined in terms of the service quality and visitor's satisfaction by Chen et al. [38]. As the most important attributes, the respondents pointed to three items included in the survey: Environmental cleanliness, ticketing process, and air conditioning. Hung-Che Wu et al. [39] conducted research concerning the attributes associated with, inter alia, service quality perceived by visitors to Macau museums. The study results based on structural equation modelling analysis and findings show four primary dimensions and 12 sub-dimensions of service quality in exanimated museums. However, Parasuraman et al. [20] indicate that SERVQUAL is used in a variety of service industries, yet its applicability to museums in post-mining areas has not been examined. The attempt at modifying SERVQUAL method and its application to assess the quality of tourist services offered by the Silesian Museum. Many industries are characterised by specific conditions, thus their functioning has a significant impact on visitors' expectations towards providers of services. The Museum was established as a result of reclamation and revitalisation the inactive "Katowice" Hard Coal Mine It is worth highlighting the specificity of the research subject—the Silesian Museum has a much wider range of services than a standard museum. With its historic infrastructure and collections of exhibits related to the industrial history of the region, the museum serves as an excellent example of enhancing the offering of services and expanding target markets. Revitalising degraded areas and protecting industrial heritage ensures perfectly with the principles of sustainable development, as a new tourist attraction is created on an already transformed land, thus protecting existing green areas. Additionally, attractive revitalisation of the post-mining areas can increase social acceptance for the extraction and processing of mineral resources.

## 3. The Characteristics of the Subject of the Study

The "Katowice" (originally "Ferdinand") Hard Coal Mine—now the Silesian Museum—is located in the Upper Silesia region in the city of Katowice (Figure 1).

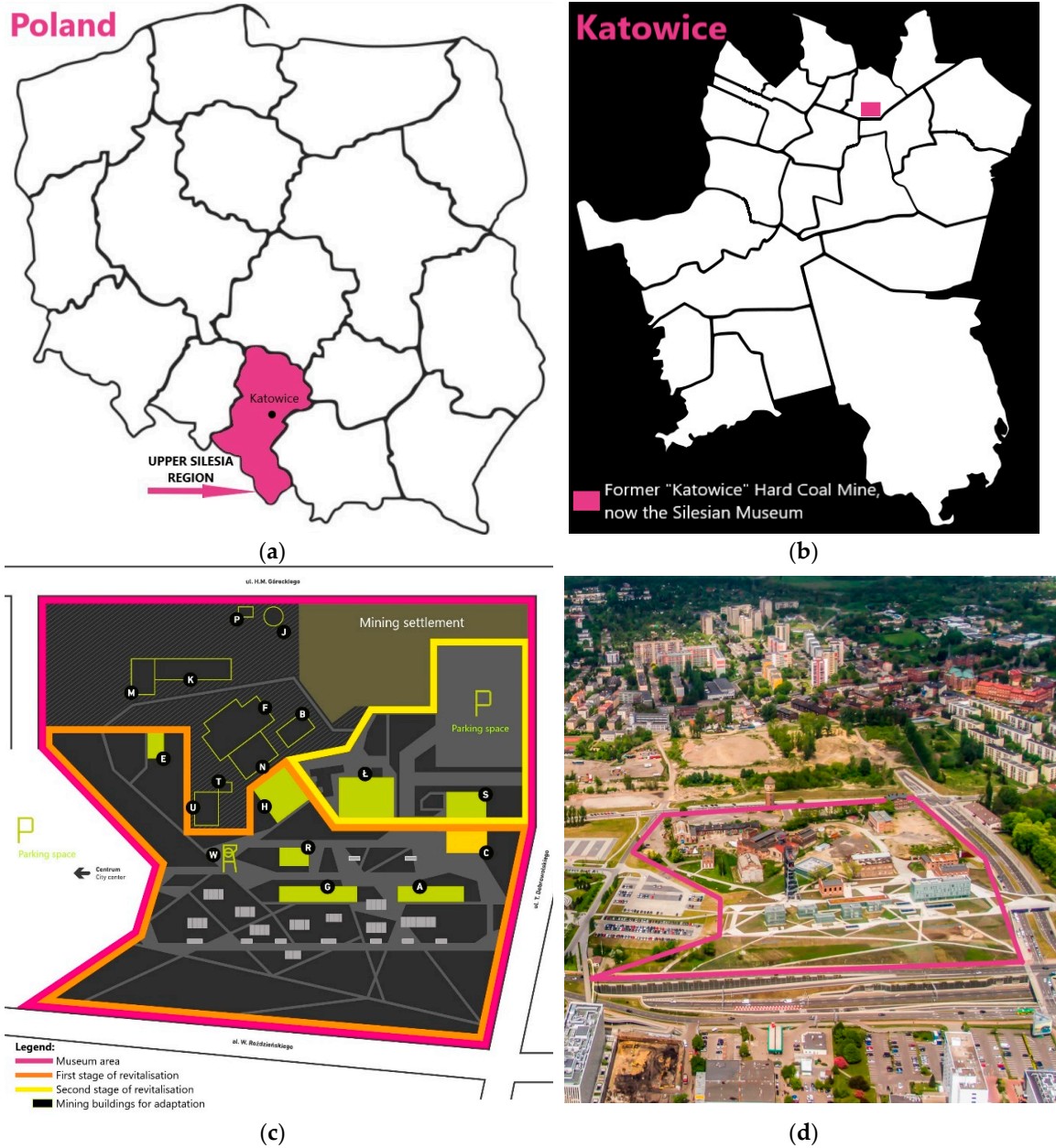

**Figure 1.** Research area location: (**a**) Location of Upper Silesia region and the city of Katowice in Poland; (**b**) Location of "Katowice" Hard Coal Mine, currently the Silesian Museum in Katowice; (**c**) Plan of the Silesian Museum with revitalisation (development) zones. Historical mining buildings: R—"Warszawa" shaft lift engine house (now a restaurant), W—"Warszawa" shaft lift tower (now an observation tower), C—Clothing warehouse (now the Centre of Polish Scenography), M—Mechanical workshop, K—Smithy (M and K now an Exhibition—the history of local industry in Upper Silesia), S—Carpentry (now a Carpentry Gallery and Stage), Ł—Bath (now a Bath Gallery), E—Electricians' workshop, B—Bartosz shaft engine room, N—Bartosz shaft building, B—Bartosz shaft engine room, F—Bartosz electric power station, T—Drillers' workshop, U—"Gwarek" baths, J—Water tower, P—Saddler's workshop. New buildings: A—Administration, café, library, auditorium, G—Main entrance, ticket windows, exhibitions, museum shop, H—Central hall, ticket window, information desk; (d) Aerial view of the Silesian Museum (as of 2016). Own source based on: (c) https://muzeumslaskie.pl/pl/architektura-i-przestrzen/ (access: 28 August 2020) (**d**) photo: K. Krzemiński.

From the 1890s, the surface buildings of the mine were erected in the nineteenth-century neo-Gothic and Art Nouveau styles. After the Mine was closed, some of facilities were destroyed or removed

by owner, while others were subjected to official conservation protection. In 2004, preparations for the adaptation of post-mining site as the new seat of the Silesian Museum began. From that moment, the history of the Mine and the Museum cannot be separated (Table 1).

**Table 1.** History of the Katowice Hard Coal Mine and Silesian Museum (source: Own study).

| History of the Mine | History of the Museum |
|---|---|
| **1823:** Commencement of the operation of "Katowice" ("Ferdinand") Hard Coal Mine | **1929:** Foundation of the Silesian Museum |
| **1999:** Closure of the "Katowice" Hard Coal Mine | **1941–1944:** Demolition of the building of the Silesian Museum by the occupier |
| | **1984:** Reinstating of the Silesian Museum and adaptation of the former hotel building into a temporary seat |
| **Common history of the Mine and Museum** | |
| **2004:** Commencement of the development process—revitalisation of the post-mining area for the new seat of the Silesian Museum | |
| **2015:** Opening of the Silesian Museum to the public (1st stage) | |
| **2017:** Completing the 2nd stage of revitalisation (adaptation of two former mining buildings and provision of infrastructure) | |
| **Plan:** 3rd stage of revitalisation (adaptation of the next nine former mining buildings) | |

The southern part of the mining area including the "Warszawa" shaft lift engine house, "Warszawa" shaft lift tower and clothing warehouse formed part of the construction of a new headquarters of the Silesian Museum. Historical facilities were transformed into a restaurant and an observation tower (Figure 2), and for exhibition purposes for the Centre of Polish Scenography. The remaining buildings, after preliminary securing, await their target functions to form an outdoors exhibition [40]. An exhibition concerning the history of industry in Upper Silesia is being prepared inside the mechanical workshop and smithy (the collection was open to the public; it is currently under reconstruction). For this purpose, artefacts related to the Polish mining industry and the Silesian steel industry are being collected (Figure 3). Among the many exhibits, a nineteenth-century steam engine that propelled two lines of rolling mills at the Baildon Steelworks can be found—the only such exhibit preserved in Poland.

New facilities were built next to historic ones, with the majority of storeys located underground, and only glazed structures visible above ground. This solution resulted from adopting an approach which involved minimising the impact on the existing urban layout of the old Mine, allowing more natural light inside exhibition spaces, while the design still took inspiration from the original function of the area: Underground mining. The new facilities form the core of the Silesian Museum. It includes six thematic galleries that present Polish art and stage design for film and theatre, as well as non-professional art (e.g., paintings by miners), sacral art and an exhibition about the history of Upper Silesia, shaped by industrialisation. There are also temporary exhibitions and conference halls. Beside exhibitions, the Museum's offer includes cultural events (e.g., open-air concerts, and once a year "The Industriada"—the Feast of the Industrial Monuments Trail) and educational events (e.g., thematic workshops). Such a combination of art and the mining history defines the target tourist audience of the museum, simultaneously attracting art lovers and industrial heritage enthusiasts. On the premises of the Silesian Museum, there is also a restaurant (in the former engine house) serving traditional regional dishes whose quality is confirmed by a certificate (it is included in the Silesian Tastes Culinary Trail).

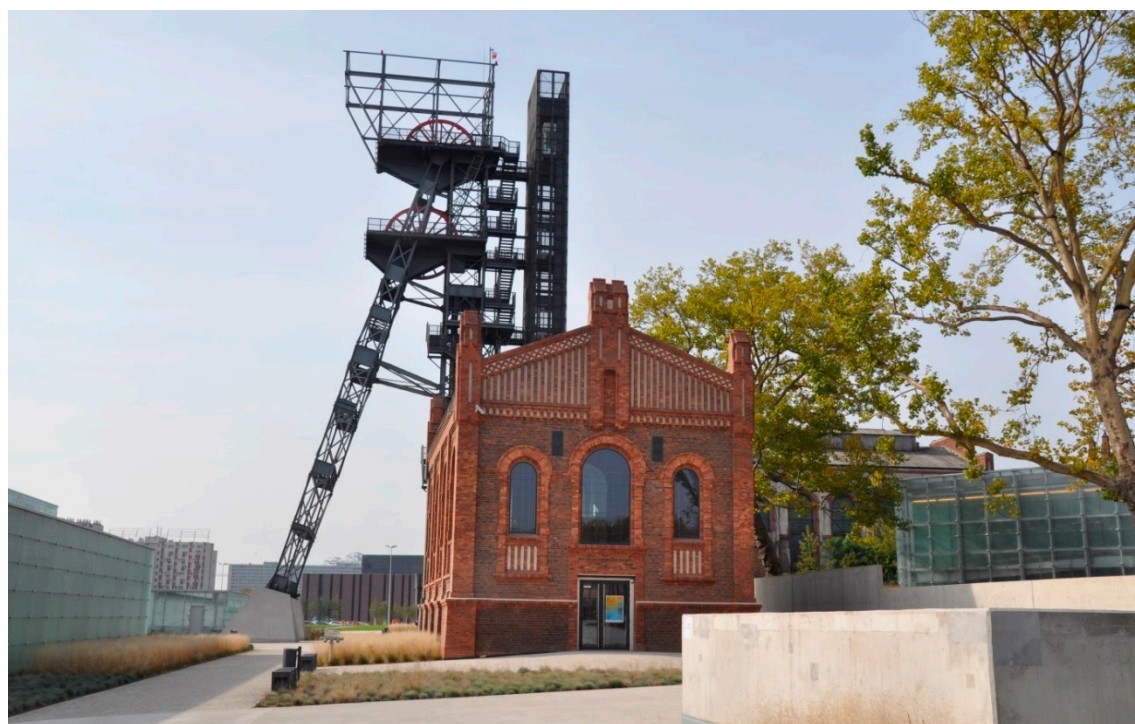

**Figure 2.** The "Warszawa" shaft lift engine house converted into a restaurant, the lift tower with a viewing platform, and the newly erected facilities (photo: A. Ostręga).

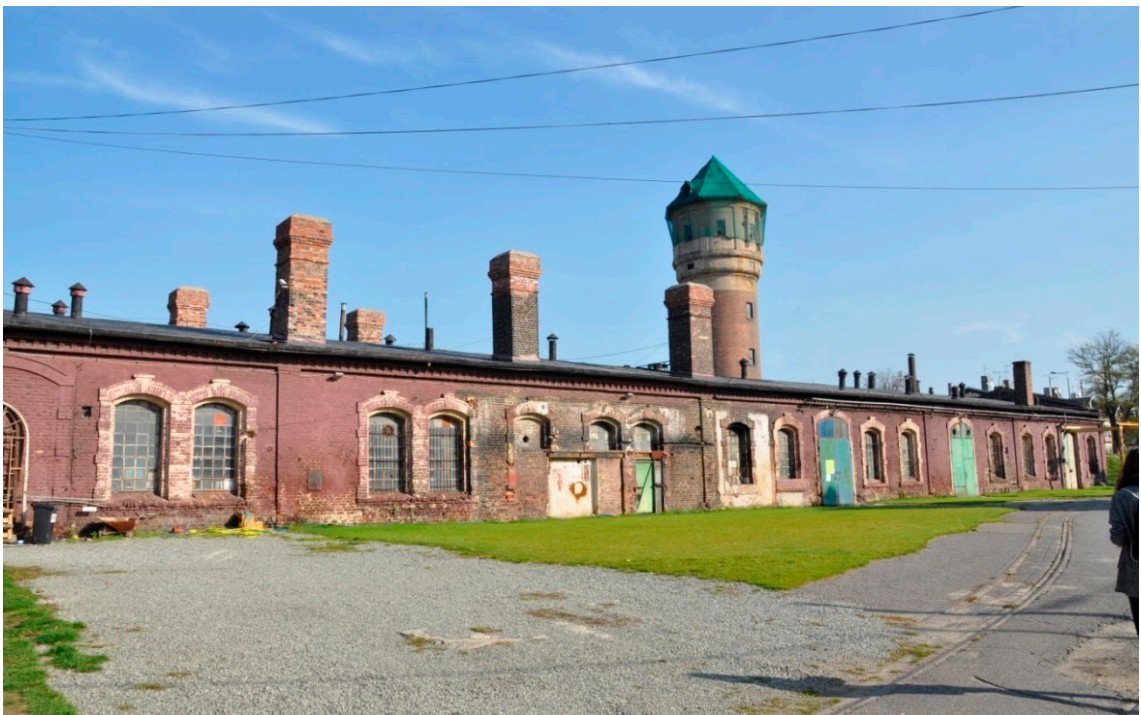

**Figure 3.** The mechanical workshop and smithy—an exhibition about the history of industry in Upper Silesia (photo: A. Ostręga).

The numbers of tourists are growing every year, and in 2019 the Museum was visited by 246,069 people, including 10,417 foreigners (for comparison, in 2016—176,071, including 6776 foreigners). The opening of the Museum to the public in 2015 was very impressive in terms of numbers—250,916 tourists were recorded then. Nearly 95% of visitors are young people (Information provided by the

Silesian Museum). The interest in the facility proves that brownfield sites, commonly associated with land degradation, can become tourist attractions after appropriate revitalisation, resulting in a positive impact on the image of the city.

Silesian Museum became part of the Industrial Monuments Route of the Silesian Voivodeship. The route is a branded tourist product of the region featuring 42 facilities related to mining and steel-working traditions, power generation, rail engineering, communications, water production, and food industry [6].

### 3.1. The SERVQUAL Method

SERVQUAL (SERvice QUALity) is the method for assessing service quality. It was developed in the 1980s by a team from the University of Miami led by Professor Parasuraman. Parasuraman's model became a starting point for theoretical and practical deliberations. The literature indicates numerous revisions of the model in an attempt to adapt it to the subject of research, based on the theoretical principles proposed by Parasuraman's team. The suggestions concerning improving the SERVQUAL method can be found in the reference books. Attempts at devising new models were made by Calabrese and Scoglio [41], Urban [42], and Baccarani et al. [43], etc. Cronin Jr and Taylor [17], contributed the most to the development of discussed method. They proposed SERVPERF as an alternative to Parasuraman's tool, and suggest resignation the element concerning expectations. Cronin Jr and Taylor highlight that making quality dependent on the perception of service is enough to determine it. SERVQUAL is often challenged as a research method [44]. Researchers pay special attention to the insufficient number of questions in the questionnaire, enclosed in five dimensions. This can result in the indicated customer experiences and needs being incomplete. On the other hand, as Urbaniak [45] emphasised, developing and extending the scope of the questionnaire can result in the clarity of the results being obscured. Even though using this method has certain benefits, it must be pointed out that there are also some limitations such as measuring the expectations of excellence, which sometimes does not exist, weak discrimination between the dimensions and gap analysis not easy to generalise in terms of other areas [46]. Moreover, the generalisability of the conducted research may be challenged due to the diversity of the profile of study participants who visit the museum, seeing as regional museums are visited rather by local population, while national ones is attract wider groups. Another limitation might be the sample selection with the desired sociodemographic profile. Furthermore, service quality is a multi-dimensional concept which may be perceived differently by study participants. Finally, the literature review emphasised that for a better understanding the analysis should be performed in terms of the gap between client's perceptions and services provided.

According to the authors, the appropriate modification and adaptation of the method to the subject of the study can make it a simple and straightforward tool to obtain answers about consumer satisfaction with the offered services.

The basis for the method is composed of the five gaps model that is sometimes referred to as the GAP model [45]:

- Gap 1: Difference between what customers expect and what managers think they expect.
- Gap 2: Difference between the management's perceptions of customer expectations and the quality specification of the service.
- Gap 3: Difference between service quality specifications and the actual quality.
- Gap 4: Difference between the quality of service provided and what is communicated to the customer about the service.
- Gap 5: Difference between a customer's expectation of the service and the perception of the experience.

The model is presented in Figure 4.

The SERVQUAL method is based on the gap 5. Therefore, it is important to demonstrate the differences between expected quality (E) and perceived quality (P). As the gap 5 is the product of

previous gaps, this gap is commonly measured. While surely a simplification, this approach yields satisfactory results [48]. A questionnaire composed of 22 items was used as a research tool. The sections of the questionnaire were based on the quality dimensions proposed by Parasuraman et al. (Table 2).

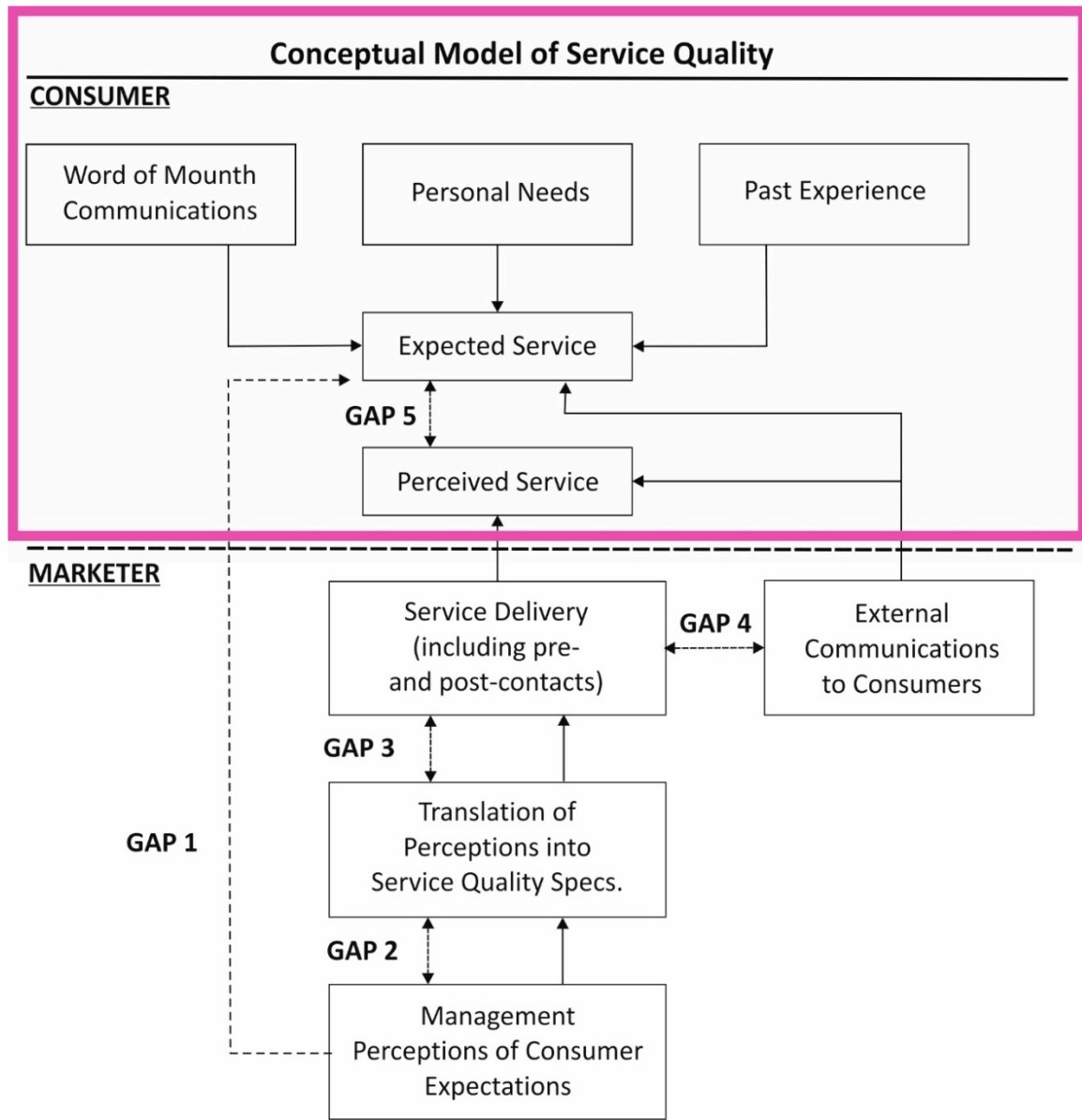

**Figure 4.** The SERVQUAL service quality model [47].

**Table 2.** Service quality dimensions in SERVQUAL [20].

| Dimension | Characteristics |
|---|---|
| Tangibles | Facilities, infrastructure, equipment, supplies, appearance of staff |
| Reliability | The ability to perform the promised service dependably and accurately |
| Responsiveness | Willingness to help customers and provide prompt service |
| Assurance | Knowledge and courtesy of employees and their ability to convey trust and confidence |
| Empathy | Caring, individualised attention the firm provides its customers |

The respondents make assessments twice—before and after visiting the museum. This made it possible to familiarise oneself with the expectations and juxtapose them with the delivered service.

A difference in their evaluations serves as a basis for demonstrating gaps in the offering. It is proposed that substantial differences in the assessments prove the presence of deficiencies in the service. The method is often expressed using the following formula [45]:

$$S = \sum (E\text{-}P), \tag{1}$$

where:

S—the degree to which consumers' expectations have been met.
E—expectations as to service quality.
P—the perceptions of service quality.

The outcome of the survey can have one of the following variants:

- Variant 1. Customers' expectations are adequate to the quality of the service delivered.
- Variant 2. Customers' expectations exceed the quality of the service delivered.
- Variant 3. Customers' expectations do not exceed the quality of the service delivered.

The methodological approach consists of the elements presented in Figure 5.

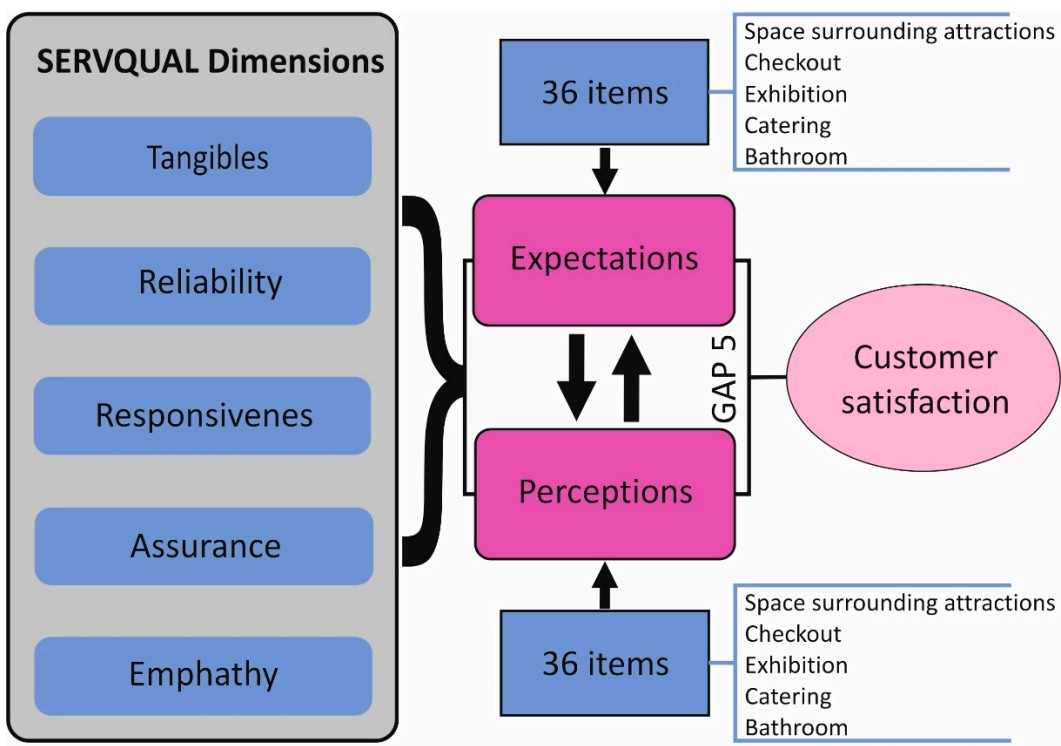

**Figure 5.** The main elements of the methodological approach used in the presented research (source: own study).

### 3.2. The Verification of the Method

The verified SERVQUAL method was checked on the basis of the example of the Silesian Museum established as a result of the revitalisation of a closed hard coal mine for cultural and tourist purposes. A form was developed on the basis of the SERVQUAL questionnaire [20]. A modified questionnaire, adapted for the specificity of the services provided by the Silesian Museum, was used. The questionnaire was prepared on the basis of the model attributes of the regional tourist product proposed by Buczak [12]. The division of the basic elements of tourist attractions was maintained, e.g., the entrance, front desk, souvenirs, exhibition, catering, and toilets [49]. Based on the main elements,

statements were chosen taking into account the dimensions proposed by the Parasuraman's team. The research tools are composed of two survey questionnaires (the expectation questionnaire and the perception questionnaire) which include 36 questions, as opposed to 22 items in the Parasuraman's model. The increased number of questionnaire items is the result of the method's verifications and striving for the goal of the method being applicable to the questionnaire concerning the subject of this paper—a mining museum. The questionnaire has been supplemented with items that allow to assess the specific aspects of a museum that is an adaptation of a brownfield site, an important part of which are exhibitions devoted to industrial heritage. This higher number of questions than in Parasuraman's model makes it possible to more comprehensively assess the expectations of the visitors. Table 3 presents items from Parasuraman et al. [20] paper along with an indication which items of the questionnaire, which are the subject of this study, were taken from the original source. The original SERVQUAL scale has been tested in terms of representatives of four sectors: a bank, a credit-card company, a firm offering appliance, repair and maintenance services. Not all items of the original questionnaire could be used in the research into a mining museum. Parasuraman's contribution was an inspiration for this research. The questionnaire items (17, 19, 21, and 22) do not correspond to the original items; however, they fit the service quality dimensions proposed in SERVQUAL (Table 2). Thus, the main advantage of the SERVQUAL method is to indicate the framework for the questionnaires and thus allow for the adaptation of specific questions to the specificity of the examined facility, service nature, and local conditions. However, the questionnaire construction on the basis of the proposed method has also certain disadvantages such as adjusting the questions to the five dimensions which can sometimes be difficult in terms of the type of conducted research.

**Table 3.** Use of the original items in the construction and modification of questions for the subject of research. Own source based on [20].

| The Original Servqual Items [20] | | The Question Number in the Proposed Original-Based Questionnaire |
|---|---|---|
| **Expectations** | **Perceptions** | |
| E1. They should have up-to-date equipment | P1. XYZ has up-to-date equipment | 5, 6, 15 |
| E2. Their physical facilities should be visually appealing | P2. XYZ's physical facilities are visually appealing | 1, 2, 3, 30 |
| E3. Their employees should be well dressed and appear neat | P3. XYZ's employees are well dressed and appear neat | 4, 31, 34 |
| E4. The appearance of the physical facilities of these firms should be in keeping with the type of services provided | P4. The appearance of the physical facilities of XYZ is in keeping with the type of services provided | 20, 23 |
| E5. When these firms promise to do something by a certain time, they should do so | P5. When XYZ promises to do something by a certain time, it does so | Not included |
| E6. When customers have problems, these firms should be sympathetic and reassuring | P6. When you have problems, XYZ is sympathetic and reassuring | 9, 10, 11, 12, 26 |
| E7. These firms should be dependable | P7. XYZ is dependable | 18, 27, 33, 35, 36 |
| E8. They should provide their services at the time they promise to do so | P8. XYZ provides its services at the time it promises to do so | 8 |

| The Original Servqual Items [20] | | The Question Number in the Proposed Original-Based Questionnaire |
|---|---|---|
| Expectations | Perceptions | |
| E9. They should keep their records accurately | P9. XYZ keeps its records accurately | Not included |
| E10. They shouldn't be expected to tell customers exactly when services will be performed | P10. XYZ does not tell customers exactly when services will be performed | 16 |
| E11. It is not realistic for customers to expect prompt service from employees of these firms | P11. You do not receive prompt service from XYZ's employees | 24 |
| E12. Their employees don't always have to be willing to help customers | P12. Employees of XYZ are not always willing to help customers | Not included |
| E13. It is okay if they are too busy to respond to customer requests promptly | P13. Employees of XYZ are too busy to respond to customer requests promptly | 32 |
| E14. Customers should be able to trust employees of these firms | P14. You can trust employees of XYZ | Not included |
| E15. Customers should be able to feel safe in their transactions with these firms' employees | P15. You feel safe in your transactions with XYZ's employees | 13, 14 |
| E16. Their employees should be polite | P16. Employees of XYZ are polite | 7 |
| E17. Their employees should get adequate support from these firms to do their jobs well | P17. Employees get adequate support from XYZ to do their jobs well | Not included |
| E18. These firms should not be expected to give customers individual attention | P18. XYZ does not give you individual attention | 24 |
| E19. Employees of these firms cannot be expected to give customers personal attention | P19. Employees of XYZ do not give you personal attention | 25 |
| E20. It is unrealistic to expect employees to know that the needs of their customers are | P20. Employees of XYZ do not know what your needs are | 28, 29 |
| E21. It is unrealistic to expect these firms to have their customers' best interests at heart | P21. XYZ does not have your best interests are heart | Not included |
| E22. They shouldn't be expected to have operating hours convenient to all their customers | P22. XYZ does not have operating hours convenient to all their customers | Not included |

The first questionnaire included statements expressing the expectations of consumers regarding services. The other covered the same statements, evaluating the quality of the delivered services. The questions were prepared according to the assumptions determining service quality: Tangibles, reliability, responsiveness, assurance, and empathy. The form is evaluated on a five-point Likert scale, commonly used in the methodology of social studies [50]. An odd-numbered scale makes it possible to give a neutral rating [51].

The respondents completed the survey twice—before and after benefiting from the service. This made it possible to familiarise oneself with the expectations and juxtapose them with the

delivered service. Table 4 presents a tool which was used. The study group was composed of AGH University students from Europe, Asia, and Poland. The conducted study was exploratory and, in addition, the sampling was purposive. The purposeful sampling method was used which is widely applied in qualitative research [52]. The survey included 30 individuals and according to statistics science this is a small sample (n ≤ 30). The research group included young adults aged 18 to 23. All study participants were students of the fields related to mining engineering, including mining engineering, mechanical engineering, and environmental engineering. Therefore, the topics related to the revitalisation of post-industrial areas are widely known to them due to the prepared theoretical introduction, which included the case study of the Silesian Museum. The respondents were asked to fill out a questionnaire before and after visiting the museum. For the dedicated study, the questionnaires were filled out in 2018 by a relatively small group, belonging to the same sociodemographic category. The obtained results and small group involvement prevented several limitations such as gender distribution.

**Table 4.** Questionnaires construction.

| Expectation Questionnaire Perception Questionnaire | | | | |
|---|---|---|---|---|
| 1 | 2 | 3 | 4 | 5 |
| strongly disagree | disagree | neutral | agree | strongly agree |
| 36 questions | | | | |
| 5 dimensions | | | | |

## 4. Results

Arithmetic average for the individual questions were calculated. The analysis suggests that the visitors' expectations were the greatest as regards the "cleanliness of restaurant space" ($\bar{x}$ = 4.53). The visitors had only slightly lower expectations concerning the "cleanliness of bathrooms" ($\bar{x}$ = 4.50). Other significant elements were the "speed of service in the restaurant space" ($\bar{x}$ = 4.37), "cleanliness of space surrounding attractions" ($\bar{x}$ = 4.33), "checkout staff friendliness" ($\bar{x}$ = 4.33), "number of bathrooms" ($\bar{x}$ = 4.33) and their "availability" ($\bar{x}$ = 4.20). Also important were "queuing time" ($\bar{x}$ = 4.10) and "information on potential risks" ($\bar{x}$ = 4.10). Furthermore, the "price of restaurant service" ($\bar{x}$ = 4.07) and "prices of souvenirs" ($\bar{x}$ = 4.07) also constituted significant factors. The exhibition and their constituents were ranked as less important—"temporary exhibition"— interesting composition, presenting the problem as a logical whole" ($\bar{x}$ = 3.97), "permanent exhibition—interesting composition, presenting the problem as a logical whole" ($\bar{x}$ = 3.83). The visitors' expectations as regards the exhibition "related to industrial heritage" were surprisingly low (x = 3.43). The visitors had the lowest expectations in respect of the "observation tower" ($\bar{x}$ = 3.30). The arithmetic average for all the items in the questionnaire was $\bar{x}$ = 3.84, with a standard deviation of δ= 0.35. This proves that the visited attractions had to meet relatively high requirements.

After visiting the Silesian Museum, the surveyed individuals completed a questionnaire concerning their evaluation of the delivered service. The items concerning the "cleanliness of bathrooms" ($\bar{x}$ = 4.83), "number of bathrooms" ($\bar{x}$ = 4.67) and "cleanliness of space surrounding attractions" ($\bar{x}$ = 4.63) were ranked the highest. Additionally, those related to the "architectural values of buildings" ($\bar{x}$ = 4.50), "permanent exhibition interesting composition, presenting the problem as a logical whole" ($\bar{x}$ = 4.5), "speed of service ( . . . ) ($\bar{x}$ = 4.50) and "availability of bathrooms" ($\bar{x}$ = 4.50) received high scores. In addition, the "marking of buildings" ($\bar{x}$ = 4.47) and "checkout staff friendliness" ($\bar{x}$ = 4.43) were appreciated by the visitors. The "observation tower" ($\bar{x}$ = 1.33) garnered the lowest scores. Due to the anonymity of the questionnaire, it was not possible to compare the responses between representatives of different nationalities. The mean score was high and amounted to $\bar{x}$ = 4.18, with a standard deviation of δ=0.54. The results of the study are presented in Table 5. The charts of the dependence of expectations

and perception of individual items in the questionnaire is included in Figure 6. Figure 7 charts of the difference between expectations average and perceptions average.

**Table 5.** Results.

| No. | Questionnaire Item | $\sum \bar{x}$(E) | $\sum \bar{x}$(P) | D=$\sum \bar{x}$(E)−$\sum \bar{x}$(P) | δ(E) | δ(P) |
|---|---|---|---|---|---|---|
| Space surrounding attractions | | | | | | |
| 1 | Architectural values of buildings | 3.73 | 4.50 | −0.77 | 1.17 | 0.78 |
| 2 | Green spaces and structural landscaping | 3.67 | 4.27 | −0.60 | 1.09 | 0.74 |
| 3 | Marking of buildings | 3.93 | 4.47 | −0.53 | 1.11 | 0.63 |
| 4 | Cleanliness | 4.33 | 4.63 | −0.30 | 0.71 | 0.56 |
| 5 | Amenities for people with disabilities | 3.43 | 4.40 | −0.97 | 1.25 | 0.97 |
| 6 | Car park | 2.90 | 4.10 | −1.20 | 1.35 | 1.12 |
| Checkout | | | | | | |
| 7 | Staff friendliness | 4.33 | 4.43 | −0.10 | 0.71 | 0.73 |
| 8 | Queuing time | 4.10 | 4.37 | −0.27 | 0.76 | 0.93 |
| 9 | Information on the tourist attraction | 3.93 | 4.23 | −0.30 | 1.08 | 0.97 |
| 10 | Information on the spatial layout of attractions | 3.70 | 4.17 | −0.47 | 0.99 | 0.95 |
| 11 | Paper maps of the complex | 3.53 | 3.90 | −0.37 | 1.14 | 1.09 |
| 12 | Information on the Industrial Monuments Route | 3.77 | 4.17 | −0.40 | 1.10 | 1.12 |
| 13 | Information on potential risks | 4.10 | 4.10 | 0.00 | 0.92 | 1.03 |
| 14 | Information on banned activities | 3.73 | 4.33 | −0.60 | 1.01 | 0.76 |
| 15 | Amenities—lift, escalator, ramp, cloakroom | 3.80 | 4.27 | −0.47 | 0.81 | 0.87 |
| 16 | Availability of souvenirs | 3.57 | 4.10 | −0.53 | 1.04 | 0.96 |
| 17 | Attractiveness of souvenirs | 3.83 | 4.10 | −0.27 | 0.95 | 0.92 |
| 18 | Prices of souvenirs | 4.07 | 4.13 | −0.07 | 0.78 | 0.97 |
| Exhibition | | | | | | |
| 19 | Observation tower ("Warszawa" shaft lift) | 3.30 | 1.33 | 1.97 | 1.21 | 1.97 |

**Table 5.** *Cont.*

| No. | Questionnaire Item | $\sum \bar{x}(E)$ | $\sum \bar{x}(P)$ | $D=\sum \bar{x}(E)-\sum \bar{x}(P)$ | $\delta(E)$ | $\delta(P)$ |
|---|---|---|---|---|---|---|
| 20 | Preservation of the industrial nature of the facility | 3.60 | 4.37 | −0.77 | 1.04 | 0.89 |
| 21 | Temporary exhibition—interesting composition, presenting the problem as a logical whole | 3.97 | 4.40 | −0.43 | 1.00 | 0.72 |
| 22 | Permanent exhibition—interesting composition, presenting the problem as a logical whole | 3.83 | 4.50 | −0.67 | 1.02 | 0.78 |
| 23 | Reference to industrial heritage | 3.43 | 4.03 | −0.60 | 1.01 | 1.13 |
| 24 | Informative exhibition staff | 3.67 | 4.20 | −0.53 | 0.96 | 0.89 |
| 25 | Exhibition which stimulates discussions | 3.53 | 4.10 | −0.57 | 0.82 | 0.96 |
| 26 | Good marking | 3.77 | 4.30 | −0.53 | 0.97 | 0.92 |
| 27 | Readability and complementarity of descriptions | 3.70 | 4.30 | −0.60 | 0.79 | 0.84 |
| Catering | | | | | | |
| 28 | Attractiveness of the menu | 3.77 | 3.87 | −0.10 | 1.07 | 1.07 |
| 29 | Vegetarian offering | 3.73 | 3.77 | −0.03 | 1.20 | 1.01 |
| 30 | Layout of dining rooms | 3.60 | 4.07 | −0.47 | 0.81 | 0.98 |
| 31 | Cleanliness | 4.53 | 4.40 | 0.13 | 0.68 | 0.81 |
| 32 | Speed of service | 4.37 | 4.50 | −0.13 | 0.76 | 0.73 |
| 33 | Price of restaurant service | 4.07 | 3.93 | 0.13 | 1.05 | 0.94 |
| Bathrooms | | | | | | |
| 34 | Cleanliness | 4.50 | 4.83 | −0.33 | 0.68 | 0.38 |
| 35 | Number | 4.33 | 4.67 | −0.33 | 0.71 | 0.61 |
| 36 | Availability | 4.20 | 4.50 | −0.30 | 0.89 | 0.86 |
| Average ($\bar{x}$) | | 3.84 | 4.18 | | | |
| Standard deviation ($\delta$) | | 0.35 | 0.54 | | | |

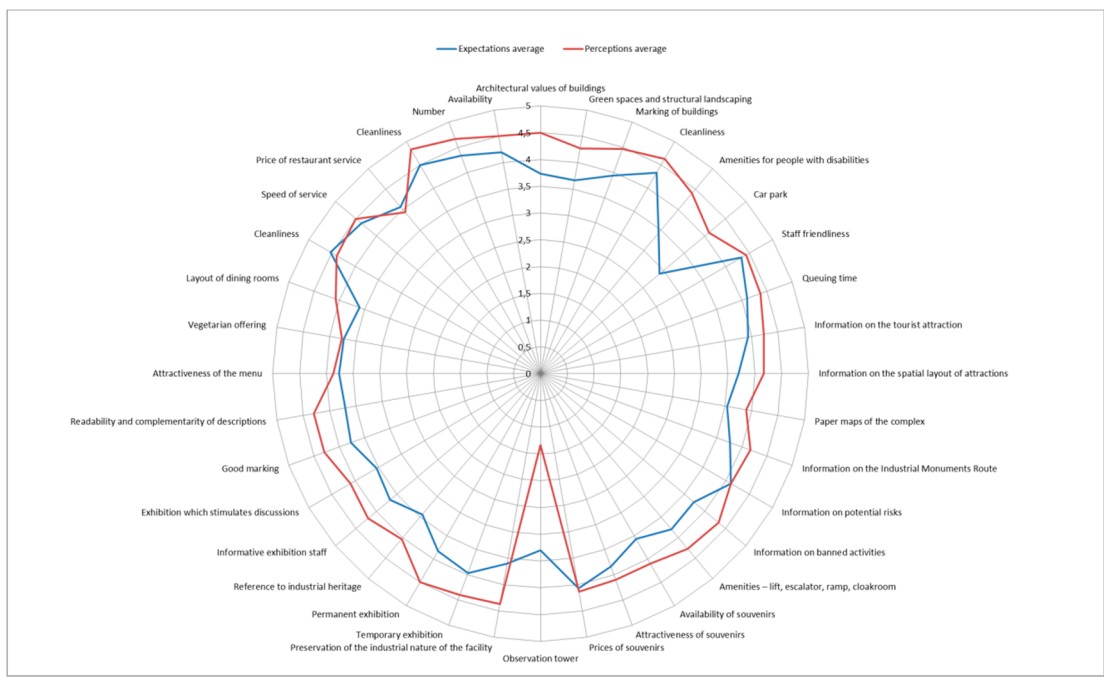

**Figure 6.** The charts of the dependence of expectations and perception of individual items in the questionnaire (source: own study).

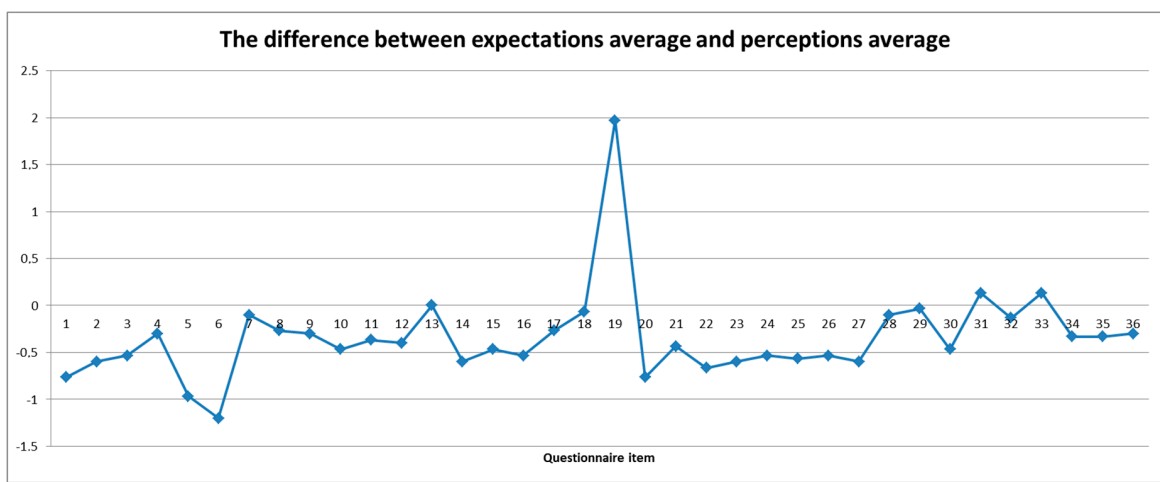

**Figure 7.** The charts of the difference between expectations average and perceptions average (source: own study).

## 5. Discussion

The analysis demonstrated that the expectations of the visitors were not met in three cases only. It suggests a very high quality of the Silesian Museum as a tourist attraction. The greatest difference between the expected and received values was found in respect to the "observation tower" (D = 1.97), with the visitors viewing it negatively. This was due to the fact that on the day of the survey this facility was unavailable for visitors. The results of the study in terms of expectations vs. perceptions demonstrate that virtually all elements differ significantly from the visitors' expectations, exceeding them (expect for "prices of souvenirs" (D = 0.07). No differences in assessment were noted only in the case of "information on potential risks" (D = 0). It is reassuring that the assessment of difference between the expectations regarding the items referring to the region's industrial heritage (architectural values of buildings, preservation of the industrial nature of the facility, permanent exhibition...) and the services delivered was significantly positive. The greatest expectations of the

visitors were associated with the "cleanliness of catering space" and the "cleanliness of bathrooms", i.e., aspects of secondary importance, which do not determine the success of a tourist attraction in the long term. The greatest difference was observed as regards "amenities for people with disabilities" (D = 0.97) and "car park" (D = 1.20).

The used scale of expectations and perceptions is a modified tool which turned out to be a unidimensional measure with a good reliability. The received results show that the adopted scale is an excellent tool which might be used to assess the magnitude of the gaps in unusual museums such as one located in a former hard coal mine. Nowacki [49], who verified the method on the example of Rogalin Palace (a National Museum), in his research also included findings indicating general satisfaction with the visit which was evaluated on the basis of four separate statements: "How do you evaluate the quality of this tourist attraction in general? Are you generally satisfied with your visit to this attraction? Would you recommend this attraction to your friends? Would you like to visit this place again in the future?". Thus, the method provides us with the possibility of verification and adaptation to the subject of research as well as the preferences and experience of the author in terms of a questionnaire. In addition, in Nowacki's study, and in the one being the subject of this article, the Likert 5-point scale was used instead of the 7-point scale as opposed to the one used in the original paper, which may also be a good guideline to follow by other authors. Such action was implemented in order to avoid making it difficult for the respondents to choose their answers and to prevent the reduction of extreme values.

The study provides the relevant information for museum's managers and policy makers, especially nowadays when most European countries are implementing changes in their museum systems in an effort to adapt to a changing reality [53]. This is of particular importance in the context of the planned expansion of the Silesian Museum through the adaptation of next historical buildings (see Figure 1 and Table 1). In addition, attention should be paid to the risks and limitations faced by museums related to online visits and hence the revenue from ticket sales, insufficient subsidies from national and regional funds which are generally dependant on the political aspects. Another key issue is the growing transformation of organisational structures of the mining industry and redevelopment funding brought about by ever-tighter restrictions. Thus, it leads to consideration of the possibilities for a wider range of sources of finance, visitor orientations and, finally, efficient management. The heterogeneous nature of mining sites and the museum's capacity to adapt to this new environment varies depending on a number of factors, including visitors' interests, perceptions, and expectations, or organisational structure and funding sources. The mining museums endowed with their specific nature may easily be likened to other more "traditional" museums since they aim to conduct commercial activity and meet financial objectives. Thus, only attempts to increase the quality of services could improve the performance of the post-industrial museums and make them attractive, especially that ones whose financial structure has impact on the innovation level. A major undertaking of the future research is to assess the SERVQUAL method applicability to other mining museums, including ones which are run by non-profit organisations and have a different level of funding.

## 6. Conclusions

The survey demonstrated that the Silesian Museum, as a tourist attraction, provides its visitors with great satisfaction. Despite a relatively small sample size (n = 30) and purposive sampling, the analysis facilitated the verification of the research tool at a brownfield site. However, the purposive sampling and small size unquestionably limit this research. Introducing the right changes to the method in relation to the subject of work can provide a useful tool to obtain information about the level of visitors' satisfaction. Therefore, the novelty of the presented research is the modification of the SERVQUAL method to the specific facility concerned with industrial heritage, as well as the use of this method to assess the quality of services provided by the museum established in the post-industrial area and offering industrial history exhibitions among its services. The implementation of projects revitalising tourist functions should also be assessed on a regular basis. Drawing the attention of

the managers of brownfield sites (revitalised to serve tourist functions) to the issues of attraction assessment and gap identification, is an important conclusion stemming from the conducted analysis. The measurement of service quality is the significant managerial tool to understand the client's needs which provides information on the weaknesses and strong points of a business. Likewise, the quality services sphere, as a subject of increased interest for both theoretical and practical considerations, should become an important part of the mining museums development. However, to provide the results, the activity in the sphere of quality research services must be implemented systematically to provide conclusions and facilitate further actions based on them. The successful attempt at verifying the SERVQUAL method for the evaluation of provided services indicates the need for further research, as the attractions and services of the Silesian Museum expand (there is such a plan), as well as in other tourist groups. The authors hope that the results of this paper will not only be used by the Silesian Museum in the process of its development, but that similar studies will also be carried out in other industrial museums or post-industrial facilities providing cultural services. The key aspect for such museums is to preserve the authenticity and emphasise the identity, therefore the evaluation of the quality of services and the exchange of experiences are extremely important.

The modified SERVQUAL method is a succinct, useful instrument for expectations and perceptions observation. The number of received responses confirmed the usefulness of the tool which might be, for future studies, replicated in other post-industrial museums with larger groups, and expanded and completed with a statistical analysis such as T-test or linear regression.

**Author Contributions:** Conceptualisation, N.K.; methodology, N.K.; validation, N.K. and A.O.; formal analysis, N.K. and A.O.; investigation, N.K.; resources, N.K. and A.O.; data curation, N.K. and A.O.; writing—original draft preparation, N.K. and A.O.; writing—review and editing, N.K. and A.O.; visualisation, A.O. All authors have read and agreed to the published version of the manuscript.

**Funding:** This research was funded by the Polish Ministry of Science and Higher Education, within the framework prescribed by statute No. 11.11.100.597.

**Conflicts of Interest:** The authors declare no conflict of interest. The funders had no role in the design of the study; in the collection, analyses, or interpretation of data; in the writing of the manuscript, or in the decision to publish the results.

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
