# Peer review of "Using SERVQUAL Method to Assess Tourist Service Quality by the Example of the Silesian Museum Established on the Post-Mining Area"

_land, doi:10.3390/land9090333_

Round 1

Reviewer 1 Report

Point 1: The document is based on a clear research question (use the SERVQUAL Method to assess the quality of the tourist service in the mining museum) and has established a clear objective. However, despite briefly mentioning the SERVQUAL methodology and giving some references to it, the authors need to put more detail on the debate, the research gap, and should make the contribution of the article to the existing literature much sharper. I think the greatest value of the work is to study the validity of the proposed methodology in the case of a mining museum (industrial tourism) whose characteristics are different from other types of museums. I even find it interesting to adapt the SERVQUAL model for analysis in any case of an industrial museum.

Point 2: The theoretical framework should be extended, and should consider more recent contributions.

Point 3: One of the main objectives of the work is to build a methodological model. This is certainly a good contribution to literature. All of this is relevant and valuable to this document, but it should be better aligned in the way the different parts relate to each other, and a better explanation of why each step is taken in relation to the objective and the question is needed. research. You should better explain how you identified the indicators. I mean, the reason why they have been selected should be better explained in the methodology.

Point 4: It is necessary to explain in more detail the sociodemographic profile of the sample. Furthermore, the sample is too small (as indicated in the conclusions), although it could not be evaluated either, since I have no data on the universe. The time period for data collection should also be indicated.

Point 5: The presentation of the case study should be expanded (add more information on the type of restaurant, how many visitors it currently has and what its evolution has been, the profile of the visitor, cultural activities ...)

Point 6: In the results, it would be necessary to expand more information and contrast it with more data.

Point 7: Insert the theoretical background and practical implications in the conclusion in a clearer way. What would be the proposals for action to follow through on the data received?
What is the usefulness of the method in the academic field? and in the professional field?

Minor issues:
Point 1: insert in the introduction the structure of the paper.
Point 2: The introduction is well thought out from the point of view that goes from the most generic to the most specific. The object of analysis is briefly contextualized and the objective is clearly defined. However, it begins by saying that tourism is one of the fastest-growing sectors of the world economy, perhaps it is out of context in the current situation.

Author Response

Attatch in .word file. 

Reviewer 2 Report

The topic is interesting. But the presentation of results is very poor. I recommend full rewriting the paper.

  1. I think the structure of the paper is a little weird. I feel 1. Introduction and 2, the aim and subject of the study should be combined. And 3. The characteristics of the subject of the study, 4. The SERVQUAL method, and 4. The verification of the method should be combined into methodology part.
  2. To set questionnaire items, authors should suggest related literature reviews. So I recommend making literature review section with comparison of existing studies.
  3. In Result and discussion, just 1 table and 3 paragraphs are existed. It is too short. And where is the discussion? I recommend making discussion section separately with difference with existing studies, political application, and limitation.
  4. I feel sample size is too small. Could you expend the sample size? Or could explain the reason why you just set 30 samples with the evidence that it is enough samples for your study.

Author Response

Attatch in .doc file

Reviewer 3 Report

This paper investigated the gaps between expectations and actual perceptions of visitors by focusing on the industrial heritage museum with the SERVQUAL method. The topic is interesting, and English is all right. Nevertheless, since the paper has many drawbacks as a scientific paper and overall research quality, I have to say that I cannot recommend this paper for publication. These are the specific drawback: the lack of a general framework for international readers, the no novelty, no hypothesis to test, short in literature review on industrial tourism, small sample size, no information of sample composition and attributes, no unique and convincing findings nor implications for the industrial tourism development. Consequently, I am strongly impressed that this paper needs an overall additional description in many crucial aspects, which makes me impressed that this paper merely applied the methodology to their students within the Polish perspective. It is a pity that the failure to clarify problems that this facility has. Therefore, I suggest the author(s) submit this paper for polish journals. For improvement, the following comments would be useful.

  1. The abstract should briefly include the data, specific findings, and implications as well.
  2. SERVQUAL is adequate to identify something needs improvement. However, I don’t understand what service quality the study facility has and which direction to improve. The author should clearly state these issues and put an empirical hypothesis to test by that method at the beginning.
  3. As mentioned above, I doubt that the author(s) don’t understand well about what issues the industrial tourism has for development, which demonstrates that literature review on industrial tourism is necessary. I strongly suggest the authors conduct that review.
  4. The data is not well described, and the small sample size collected only from students, which has a sample bias. The author(s) should mention the attributes of the samples. The sample size 30 is too small to generalize the results, so I highly recommend increasing the sample size of nearly 100 respondents at least. Still, if you use only student samples, you should clearly mention sample bias.
  5. On the questionnaire survey, the author(s) should mention why you increase the 36 questions from orthodox 22 questions. Further, you should mention when and how the survey was conducted actually. AGH should be explained what it means.
  6. The results were not convincing and no novel findings due to the lack of hypothesis as mentioned above.
  7. The conclusion should mention what was found newly, which I can not see. Further, implications and the limits of this study are also needed to be added in the final part of the conclusion. These aspects are the must for a scientific paper, but not available in this paper.

Author Response

Attatch in .doc file

Round 2

Reviewer 1 Report

The authors have successfully managed making major changes in all parts of the paper (introduction, theory, methodology, results and conclusions), and it has improved considerably. 

Author Response

We would like to thank you for the valuable comments. with kind regards, Authors. 

Reviewer 2 Report

The paper is revised a lot. But there are still weird points.

  1. I feel that materials and methodology contents (3. The characteristics of the subject of the study, 4. The Servqual method, 5. The verification of the method) should be combined. I think two subsection (materials and methodology) recommended in one section.
  2. Map and table of the history will be helpful to understand the sites.
  3. Methodology parts are too long. Please show the key points. And diagram about methodology process will be helpful to understand.
  4. Please explain more about result and making some graphs will be beneficial for understand the statistics.
  5. Discussion section is needed separately for comparison with existing studies, political application, limitation and future studies.

Author Response

The author's reply to the following review is attached as a word file. We would like to thank you for the valuable comments. 

Reviewer 3 Report

I understand that the revision was made a certain extent. Nevertheless, the initial crucial comments were not fully addressed. I mentioned again the left point that still needs to be addressed properly.

  1. The authors said the new addition in the evaluation questionnaire is a novel point. However, how this addition was evaluated in comparison with orthodox one. The advantage and disadvantage should be clearly stated in the text. The author simply mentions the merits of this SERVQUAL methodology in the results and conclusion, but you need to be more specific points on the advantages and disadvantages resulting from your “novel analysis”, which should be more specifically clarified.
  2. Related to the first comment above, the author should indicate newly added questionnaire items in Table 3 in comparison with the orthodox one.
  3. On the data, I mentioned the two possible problems in the dirt comments, the one is the small sample size, which you justified in the response report. It is all right but if so, you need also add this justification in the text as well.

On the 2nd issue of the data, you did not address well. I still doubt that the students' sample has data bias in terms of nationality or their evaluation. This is because this museum is not specific for college students, but wide open to ordinary people with various backgrounds and generations, which means that wider sample collection is needed for proper evaluation. To address these issues, a cross-tabulation of evaluation results between e.g., European and Asian nationalities would be suggested. If there is no difference, it is okay. But if you notice any differences between the nationality, it should be noted in the text.

Also, at the end of the conclusion, the limitation of this study should be noted such as that a wider range of sample should be considered.

   The last one, when the data was collected in the survey year. Specifically, what month?

4.On the minor point, Figure 3 is vague, which need better visibility.

Author Response

(The authors gave the same response as above.)
